# L-Alanine Prototrophic Suppressors Emerge from L-Alanine Auxotroph through Stress-Induced Mutagenesis in *Escherichia coli*

**DOI:** 10.3390/microorganisms9030472

**Published:** 2021-02-25

**Authors:** Harutaka Mishima, Hirokazu Watanabe, Kei Uchigasaki, So Shimoda, Shota Seki, Toshitaka Kumagai, Tomonori Nochi, Tasuke Ando, Hiroshi Yoneyama

**Affiliations:** 1Laboratory of Animal Microbiology, Department of Microbial Biotechnology, Graduate School of Agricultural Science, Tohoku University, 468-1, Aramaki Aza Aoba, Aoba-ku, Sendai 980-8572, Japan; 3island.1024@gmail.com (H.M.); sendtohimtwo@yahoo.co.jp (H.W.); kei@uchigasaki.com (K.U.); so.shimoda.t7@dc.tohoku.ac.jp (S.S.); jjbgobfnanaz@yahoo.co.jp (S.S.); tasuke.ando.d4@tohoku.ac.jp (T.A.); 2Fermlab Inc., 4-3-1-913, Shirakawa, Koto-ku, Tokyo 135-0021, Japan; tk@fermlab.com; 3Laboratory of Functional Morphology, Department of Animal Biology, Graduate School of Agricultural Science, Tohoku University, 468-1, Aramaki Aza Aoba, Aoba-ku, Sendai 980-8572, Japan; tomonori.nochi.a5@tohoku.ac.jp

**Keywords:** L-alanine, starvation, stress-induced mutagenesis, auxotrophic mutant, prototrophic mutant, *Escherichia coli*

## Abstract

In *Escherichia coli*, L-alanine is synthesized by three isozymes: YfbQ, YfdZ, and AvtA. When an *E. coli* L-alanine auxotrophic isogenic mutant lacking the three isozymes was grown on L-alanine-deficient minimal agar medium, L-alanine prototrophic mutants emerged considerably more frequently than by spontaneous mutation; the emergence frequency increased over time, and, in an L-alanine-supplemented minimal medium, correlated inversely with L-alanine concentration, indicating that the mutants were derived through stress-induced mutagenesis. Whole-genome analysis of 40 independent L-alanine prototrophic mutants identified 16 and 18 clones harboring point mutation(s) in pyruvate dehydrogenase complex and phosphotransacetylase-acetate kinase pathway, which respectively produce acetyl coenzyme A and acetate from pyruvate. When two point mutations identified in L-alanine prototrophic mutants, in *pta* (D656A) and *aceE* (G147D), were individually introduced into the original L-alanine auxotroph, the isogenic mutants exhibited almost identical growth recovery as the respective cognate mutants. Each original- and isogenic-clone pair carrying the *pta* or *aceE* mutation showed extremely low phosphotransacetylase or pyruvate dehydrogenase activity, respectively. Lastly, extracellularly-added pyruvate, which dose-dependently supported L-alanine auxotroph growth, relieved the L-alanine starvation stress, preventing the emergence of L-alanine prototrophic mutants. Thus, L-alanine starvation-provoked stress-induced mutagenesis in the L-alanine auxotroph could lead to intracellular pyruvate increase, which eventually induces L-alanine prototrophy.

## 1. Introduction

Bacteria are continuously exposed to environmental stresses that restrict their growth, such as nutritional deprivation and exposure to xenobiotics, but bacteria are able to survive because of a versatile, pliable, and elaborate metabolic network provided by the evolutionary process. To overcome the harmful conditions caused by the blockage of specific metabolic pathways, one strategy employed is the use of isozymes that complement a blocked enzyme [1]. In terms of pathways leading to amino acid biosynthesis, all of these pathways, except that for L-alanine biosynthesis, were revealed in the mid-to-late twentieth century through extensive microbial genetics and biochemical studies [2,3], which showed that isozymes are involved in the generation of aromatic amino acids and aspartate family amino acids [4,5]. Recently, L-alanine has also been shown to be synthesized by three isozymes—AvtA, YfbQ, and YfdZ—that use pyruvate as a substrate and catalyze aminotransferase activity [6,7].

An alternative to the use of isozymes is the complementation of damaged metabolic pathways by secondary enzyme activities, a phenomenon called “multicopy suppression” [8,9], which has been documented to occur in both catabolic pathways [10,11,12] and anabolic pathways [13] affected by mutations. As examples related to catabolic pathways, a cryptic β-galactosidase, Ebg, suppresses *lacZ* mutants [12], and fucose metabolic enzymes can be recruited for the catabolism of L-1,2-propanediol [10] and D-arabinose [11]. Conversely, in terms of the anabolic pathway, a D-alanine auxotroph produced by double-knockout mutation of the genes *alr* and *dadX* gains the ability to synthesize D-alanine through a suppressor mutation that results in high expression of *metC* [13]; this finding clearly implies that the cystathionine β-lyase MetC shows promiscuous activity and generates D-alanine when expressed at a high level. Moreover, by using a shotgun-cloning experiment, we previously found that, besides D-alanine, *serC* on a multicopy plasmid can suppress the growth defect of an *Escherichia coli* L-alanine auxotroph lacking the aforementioned three isozymes, but a single chromosomal copy of *serC* cannot [6]; our results demonstrated that SerC exhibits promiscuous activity and generates L-alanine. Furthermore, another study reported that other transaminase genes also suppress the growth defect of the L-alanine auxotroph, either efficiently (*argD* and *astC*) or partially (*aspC*, *gabT*, *puuE*, *tyrB*, and *ygiG*) [7]. These findings indicate that bacteria can respond to nutritional stress (i.e., to starvation) in a more adaptable manner than previously recognized.

During the course of our previous study, in which the L-alanine auxotrophic mutant was isolated using a traditional microbial genetics approach, we found that L-alanine prototrophic suppressor mutants appeared at a frequency higher than that through spontaneous mutation. This unexpected phenomenon poses two fundamental questions: (1) What mutation(s) conferred the L-alanine auxotroph the prototrophic phenotype? (2) Were the suppressor mutants generated due to stress-induced mutagenesis? Here, we analyzed the genome sequence of independently isolated suppressor mutants to identify the mutations that result in the L-alanine prototrophy and the response of the L-alanine auxotroph toward various conditions that relieve L-alanine starvation.

## 2. Materials and Methods

### 2.1. Bacterial Strains, Plasmids, and Culture Conditions

The *E. coli* strains and plasmids used in this study are listed in Table 1. Bacterial cells were cultured at 37 °C in Luria-Bertani (LB) medium containing 1% tryptone, 0.5% yeast extract, and 0.5% NaCl (pH 7.2) or minimal medium containing 22 mM glucose, 7.5 mM (NH_4_)_2_SO_4_, 1.7 mM MgSO_4_, 7 mM K_2_SO_4_, 22 mM NaCl, and 100 mM sodium phosphate (pH 7.1) [14]. When necessary, kanamycin (KM, 6.25 μg/mL), gentamicin (GM, 6.25 μg/mL), and chloramphenicol (CP, 25 μg/mL) were added to the medium. Growth was monitored by measuring the absorbance at 660 nm (A_660_).

### 2.2. Isolation of L-Alanine Prototrophic Clones

*E. coli* L-alanine auxotrophic strain HYE032 cells (10^6^–10^9^) were placed on minimal agar medium containing 6.25 μg/mL KM and 6.25 μg/mL GM and grown at 37 °C. At 2, 3, or 4 days after incubation, newly appeared colonies were purified in LB agar medium containing 6.25 μg/mL KM and 6.25 μg/mL GM, and the phenotypes of the resulting clones were determined on minimal agar medium supplemented with or without 20 μg/mL L-alanine.

### 2.3. Frequencies of Mutants Grown in Minimal Medium with or without L-Alanine Supplementation

*E. coli* HYE032 was grown overnight in an LB medium containing 6.25 μg/mL KM and 6.25 μg/mL GM, and the cells were collected through centrifugation (15,000× *g*, 23 °C, 5 min), washed thrice with 0.85% NaCl, and resuspended in the original volume of the same solution. We inoculated the resulting cells (5 μL) into a minimal liquid medium (5 mL) containing 6.25 μg/mL KM, 6.25 μg/mL GM, and L-alanine (100, 50, 10, 1, or 0.1 μg/mL) and cultured the cells until the late-log phase, and then counted the cells that grew on minimal agar medium containing 6.25 μg/mL KM and 6.25 μg/mL GM after incubation for 2 days at 37 °C. The frequencies of the mutants that appeared on minimal agar medium were calculated by normalizing against the total cell count obtained with LB agar medium containing 6.25 μg/mL KM and 6.25 μg/mL GM.

### 2.4. DNA Sequencing and Annotation of Mutations

Chromosomal DNA of L-alanine suppressor mutants was extracted using phenol and chloroform-isoamyl alcohol (24:1), as described [16], and after shearing the isolated DNA, a fragment library was prepared using a bar-coding protocol, according to the manufacturer’s instructions (Thermo Fisher Scientific, Waltham, MA, USA). Subsequently, genome sequencing was performed using a SOLiD 5500XL sequencer. All sequence reads obtained were mapped to the reference genome of *E. coli* W3110 (Accession Number NC_007779) by using bowtie2 [17] and samtools [18]. Candidate mutations (point mutations and InDels) as differences from the reference genome sequence were identified by FreeBayes [19] and filtered using vcftools [20]. Lastly, all candidate mutations were verified by using the dideoxy chain-termination sequencing method [21] with the primers listed in Appendix A.

### 2.5. Construction of Isogenic Mutants

To construct HYE032 isogenic mutants carrying respectively the point mutation of L-alanine prototrophic suppressors #8-2 and #13-1, we first amplified the DNA fragments harboring the point mutation of each suppressor clone by performing PCR with specific primers (Table 2) and the chromosomal DNA of #8-2 and #13-1, respectively, as the template. The resulting fragments were digested with *Xba*I and cloned into the *Xba*I site of pTH18cs1 [15], which generated pTH8-2 and pTH13-1, whose mutations were derived from #8-2 and #13-1, respectively. After the transformation of pTH8-2 and pTH13-1 into HYE032, the resulting transformants were grown in LB medium containing 12.5 μg/mL KM, 12.5 μg/mL GM, and 25 μg/mL CP at 42 °C overnight, and then integrants were selected on LB agar medium containing 12.5 μg/mL KM, 12.5 μg/mL GM, and 25 μg/mL CP; subsequently, the isogenic mutants, #8-2M and #13-1M, respectively, were obtained by selecting CP-susceptible clones.

### 2.6. Crude Cell Extract Preparation

*E. coli* cells were grown in LB medium containing appropriate antibiotics to the mid-log phase (A_660_ ≅ 0.6), collected through centrifugation (15,000× *g*, 4 °C, 5 min), washed thrice with ice-cold 50 mM potassium phosphate buffer (pH 8.0) or 50 mM Tris-HCl (pH 8.0), and then resuspended in the original volume of the same solution. The collected cells were disrupted by applying 5 cycles of 15-s sonic oscillation with 45-s intermittent cooling in an ice bath by using a Bioruptor (UCD-250, CosmoBio Co., Tokyo, Japan). The supernatant obtained through centrifugation (15,000× *g*, 4 °C, 15 min) was used for assaying enzyme activity.

### 2.7. Enzyme Activity Assay

Pyruvate dehydrogenase (PDH) activity was determined spectrophotometrically by monitoring NADH formation at 340 nm (ε_340_ = 6220 L/mol/cm) at 25 °C [22]. The reaction mixtures contained 50 mM potassium phosphate (pH 8.0), 0.2 mM thiamine pyrophosphate, 0.1 mM coenzyme A, 1 mM MgCl_2_, 0.3 mM dithiothreitol, 2.5 mM NAD^+^, 100 μg/mL bovine serum albumin, and crude cell extracts. The reaction was initiated by adding pyruvate to a final concentration of 5 mM.

Phosphotransacetylase (PTA) activity was determined spectrophotometrically by monitoring thioester bond formation at 233 nm (ε_233_ = 5550 L/mol/cm) at 25 °C [23]. The reaction mixtures contained 50 mM Tris-HCl (pH 8.0), 20 mM KCl, 0.2 mM coenzyme A, 2 mM dithiothreitol, and crude cell extracts. The reaction was initiated by adding acetyl phosphate to a final concentration of 10 mM.

### 2.8. Protein Concentration Measurement

Protein concentrations were determined by using the Lowry method with bovine serum albumin as a standard [24].

### 2.9. Statistics

Statistical analyses involving one-way ANOVA with the Kruskal-Wallis test (Figure 1a,b, Figure 5a,b and Figure 6) and t-test (Figure 6) were performed using Prism version 7 (GraphPad Software, San Diego, CA, USA).

## 3. Results

### 3.1. L-Alanine Prototrophs Emerge from L-Alanine Auxotroph through Suppressor Mutation

When the L-alanine auxotroph HYE032 [6] was continuously incubated in a minimal liquid medium for >2 days, bacterial growth was reproducibly observed in this medium without L-alanine Appendix A. To characterize the precise growth behavior of the cells that grew, the clones that proliferated in the liquid medium were purified on LB agar medium and the resulting single clones were cultured in the same minimal liquid medium. All clones started to grow considerably faster than their parental strain, HYE032, and showed a growth pattern similar to that of the wild-type strain W3110 (Appendix A). Because the L-alanine auxotroph HYE032 was constructed through homologous recombination of three genes (*yfbQ*, *yfdZ*, and *avtA*) encoding the major L-alanine synthesizing enzymes, the clones are highly unlikely to have emerged due to a back mutation in one of these genes. Thus, a suppressor mutation(s) generated the obtained clones exhibiting L-alanine prototrophy.

### 3.2. L-Alanine Prototrophic Suppressor Mutants are Generated through Stress-Induced Mutagenesis

Because the L-alanine prototrophic suppressors appeared at a frequency substantially higher than that expected for mutants generated by a spontaneous mutation, we sought to quantify the suppressors that emerged during the course of incubation (2–4 days) on minimal agar medium; the number of suppressors that appeared increased in an incubation time-dependent manner (Figure 1a), which suggested that the appearance of the suppressors was caused by mutagenesis induced by stress (here, L-alanine starvation). In this scenario, removal of the stress would be expected to prevent the appearance of the suppressor mutants, and we tested this by measuring the frequency of L-alanine prototrophic suppressors in a population of late-log-phase cells grown in a minimal liquid medium; the frequencies of appearance of the suppressor mutants were inversely related to the concentration of L-alanine added to the minimal medium (Figure 1b). These results clearly indicate that the obtained L-alanine prototrophic suppressors were generated through mutagenesis induced by stress (L-alanine starvation).

### 3.3. L-Alanine Prototrophic Suppressors Show Distinct Levels of Growth Recovery

The L-alanine prototrophic suppressors obtained in minimal liquid medium showed an overall similar pattern of growth recovery (Appendix A); this supported the possibility that the suppressors were derived from a single mutation event that generated the offspring from the original mutant HYE032, meaning that most of these suppressors could represent the same clone. Hence, they were supposed to show similar patterns of growth. We thus selected L-alanine prototrophic suppressors on minimal agar medium without L-alanine supplementation to obtain distinct mutants, i.e., genetically independent clones. After isolating several individual suppressor mutants, we measured their growth in a minimal liquid medium and observed varying degrees of growth recovery (Figure 2). Intriguingly, the suppressors obtained after 2, 3, or 4 days of incubation showed slightly dissimilar patterns of growth recovery (Appendix A); specifically, the suppressors obtained after 4-day incubation exhibited slower growth initiation than did the clones obtained within 3 days of incubation. The results suggested that individual suppressor clones, isolated using minimal agar medium carried distinct mutation(s), which contributed to the varying degrees of growth suppression of the individual clones.

### 3.4. Factors Supporting Growth of L-Alanine Auxotroph in Minimal Medium

The aforementioned results were interpreted to imply that a suppressor mutation caused a metabolic-flow change that ultimately led to L-alanine synthesis, which was achieved by an enzyme(s) other than the three major alanine synthesizing enzymes, YfbQ, YfdZ, and AvtA [6,7]. Notably, the phenomenon of multicopy suppression was observed in the case of the L-alanine auxotroph: HYE032 cells harboring a multicopy plasmid carrying *serC*, which encodes 3-phosphoserine aminotransferase, grew on minimal medium without L-alanine supplementation [6], whereas the host HYE032 cells harboring a single copy of wild-type *serC* in the genome could not grow on the same medium [6]. The results suggested that highly expressed SerC could use (or recognize) pyruvate as a minor substrate, which would then result in the provision of L-alanine to support the growth of the L-alanine auxotroph. Therefore, we assessed the impact of supplementation of pyruvate, an immediate precursor ketoacid of L-alanine, and various other amino acids, including L-glutamic acid, an amino donor of alanine aminotransferases. Pyruvate supported the growth of HYE032 cells in a concentration-dependent manner (Figure 3), and L-glutamic acid also supported the growth to a certain extent (Appendix A). These results implied that conditions that cause an increase in the levels of intracellular substrates for L-alanine synthesis, particularly pyruvate, enabled the L-alanine auxotroph to grow in minimal medium.

### 3.5. Identification of Suppressor Mutations that Cause L-Alanine Auxotroph to Grow without L-Alanine Supplementation

The results described in the preceding subsection led us to infer that the suppressor mutants isolated using L-alanine-lacking minimal medium potentially carry mutations that lead to the generation of pyruvate by inducing a metabolic change. To identify the mutations that could cause this phenotypic change, we performed whole-genome sequencing of the suppressor mutants and identified point mutations in 40 individual mutants (Table 3). All ambiguous mutations detected using a SOLiD 5500XL sequencer were verified using the Sanger method [21], and we ultimately found that 30 clones carried a mutation(s) in a single gene (Appendix A). Interestingly, among these clones, suppressor clone #39-2 harbored two distinct point mutations in *aceE*, a missense mutation (A206V) and a silent mutation (V290). Furthermore, nine suppressor clones concurrently carried a point mutation in two genes, and one clone carried a point mutation in three genes (Appendix A). The finding that multiple point mutations exist in a single clone agrees with the notion that L-alanine suppressors appeared through stress-induced mutagenesis and not spontaneous mutation.

Intriguingly, 17 clones carried a point mutation in *pta* and one clone in *ackA*; both genes are part of an acetate-synthesizing pathway, the PTA-ACKA pathway (Table 3 and Appendix A). Moreover, 11, two, and three clones carried point mutations in *aceE*, *aceF*, and *lpd*, respectively, which encode subunits of the multi-component enzyme PDH. Overall, 34/40 clones of L-alanine suppressors (85%) harbored mutations in metabolic pathways downstream from pyruvate that enters the TCA cycle and the PTA-ACKA pathway (Table 3 and Appendix A).

### 3.6. Characterization of Mutations Identified in pta and aceE

We hypothesized that the individual mutations identified here represent a major cause of the appearance of L-alanine suppressors. To verify this, we selected two suppressor mutants as model systems: #8-2, carrying a point mutation (*pta*, D656A) found in 14 independent clones; and #13-1, carrying a point mutation (*aceE*, G147D) found in two independent clones. We singly introduced the point mutations D656A and G147D into the parent L-alanine auxotroph HYE032 through homologous recombination to obtain clones #8-2M and #13-1M, respectively. The generated isogenic mutants showed growth recovery in minimal medium to a level similar to that observed in their original suppressor mutants, #8-2 and #13-1, respectively (Figure 4), which indicated clearly that the individual point mutations caused the suppression phenotype of the mutant clones.

Next, we determined the enzyme activities of PTA and PDH in the suppressor mutants and their isogenic mutants to assess the biochemical background for the suppression phenotype. As expected, the *pta* suppressor (#8-2) and its isogenic mutant (#8-2M) derived from the L-alanine auxotroph showed negligible PTA activity (Figure 5). Similarly, the *aceE* suppressor (#13-1) and its isogenic mutant (#13-1M) showed extremely low PDH activity as compared with their parental strain (HYE032) and the wild-type strain (W3110) (Figure 5). These results implied that the metabolic flow at each of these enzymatic reaction steps could be restricted, which would result in pyruvate accumulation in cells; this change in metabolic flow, in turn, could produce a shift toward L-alanine generation due to increased levels of pyruvate, the immediate precursor of L-alanine. Both enzyme activities in the L-alanine auxotroph HYE032 were comparable to those in its parental strain (W3110), which indicated that blockage of L-alanine formation itself did not influence PTA and PDH activities.

### 3.7. Impact of Extracellular Pyruvate and Anaerobic Culture Conditions on Emergence of L-Alanine Prototrophic Suppressors

The results in the preceding subsection indicated that the mutations that led to an increase in intracellular pyruvate levels represented the underlying mutations in the L-alanine prototrophic suppressors, and that a metabolic stress—here, L-alanine starvation—induced the suppressor mutations. Thus, we hypothesized that changes in intracellular levels of pyruvate could, conversely, affect the appearance of L-alanine prototrophic suppressors, and to test this, we determined the emergence frequency of L-alanine prototrophic suppressor mutants after culturing HYE032 cells in minimal liquid medium supplemented with pyruvate (Figure 6): L-alanine prototrophic suppressors appeared in the population of late-log-phase cells with decreasing frequency as the amount of extracellularly added pyruvate was increased. Because intracellular levels of pyruvate are related to extracellular pyruvate concentrations [25], the result implied that L-alanine starvation stress is relieved by pyruvate supplementation in the medium, which leads to low frequencies of appearance of L-alanine prototrophic suppressors.

To further substantiate our finding, we determined the emergence frequencies of L-alanine suppressor mutants after culturing HYE032 cells anaerobically, a condition widely recognized to increase metabolic flow through the glycolysis pathway leading to pyruvate generation [26,27]. As per our expectation, the frequencies of L-alanine prototrophic suppressor mutants were markedly decreased as compared to those observed under aerobic culture conditions with no L-alanine supplementation (Figure 6), which clearly indicated that the L-alanine starvation stress exerted on the L-alanine auxotroph HYE032 was relieved by intrinsically synthesized pyruvate.

## 4. Discussion

In this study, we found that L-alanine prototrophic suppressor mutants emerged due to mutagenesis induced by stress (in this case, L-alanine starvation); our conclusion here is based on the following results: (1) L-alanine prototrophic mutants appeared under nonlethal selection conditions in a time-dependent manner (Figure 1a); (2) the frequencies of appearance of the suppressor mutants were higher than that due to spontaneous mutation, which agrees well with other nonlethal selection systems involving the use of lactose and amino acid auxotrophs [28,29,30]; and (3) the frequencies of L-alanine prototrophic suppressors in the late-log-phase population were inversely related to the L-alanine levels present in minimal liquid medium (Figure 1b).

We propose that the suppressor mutants are generated through the following steps: (1) perception of the environmental stress (L-alanine starvation), (2) activation of an intracellular signaling cascade that currently remains unelucidated, (3) involvement of the nucleic acid metabolic process that leads to a mutation(s), and (4) L-alanine biosynthesis that supports the growth of the L-alanine auxotroph HYE032 in the absence of L-alanine supplementation (Figure 7). Regarding Step 4, the three genes encoding the isozymes AvtA, YfbQ, and YfdZ, which synthesize L-alanine, were knocked out using a gene-replacement technique, and thus reversion of each respective L-alanine-generating gene product theoretically cannot occur. Accordingly, the L-alanine prototrophic suppressor mutants could harbor mutations in a gene(s) other than *avtA*, *yfbQ*, and *yfdZ*. The results of whole-genome sequencing analysis of the obtained suppressors confirmed this view (Table 3 and Appendix A).

The most physiologically relevant result presented here is that 18 and 16 suppressor mutants were found to carry mutations in genes that function in the metabolic steps immediately downstream of pyruvate, in the PTA-ACKA pathway and PDH complex, respectively (Table 3 and Appendix A). This result agrees well with that of an earlier study in which multicopy suppression in an L-alanine auxotroph was observed with *serC* [6], the gene encoding the enzyme 3-phosphoserine aminotransferase involved in L-serine biosynthesis; this finding implies that pyruvate could be a minor substrate of SerC in the generation of L-alanine [6]. Accordingly, a mutation(s) that leads to an increase in the intracellular level of pyruvate is highly likely to occur in a suppressor mutant that carries a single copy of wild-type *serC* in the genome. This scenario is verified by the findings that (1) isogenic mutants derived from the L-alanine auxotroph HYE032 harboring the *pta* mutation (clone #8-2) or *aceE* mutation (clone #13-1) show L-alanine prototrophy (Figure 4), and (2) both isogenic mutants exhibit markedly low activity of the respective enzymes (Figure 5). The aforementioned results agree closely with the results of previous metabolic engineering studies, in which genetic manipulation leading to elimination or downregulation of PDH activity and/or the acetate metabolic pathway (*pta* and *ackA*) promoted pyruvate production [31,32,33].

In addition to the mutation-carrying genes of the PDH complex (*aceE*, *aceF*, and *lpd*) and the PTA-ACKA pathway (*pta* and *ackA*), 14 genes were found to harbor a point mutation in the L-alanine prototrophic suppressors (Table 3 and Appendix A). Notably, among these mutations, the same mutation in *cyaA* (C1957T, which produces the R653C amino acid change in the encoded adenylate cyclase) was found in four suppressor mutants: #10-1, #44-1, #64-1, and #68-1; the mutants #44-1 and #64-1 carried only the *cyaA* mutation, whereas #10-1 and #68-1 harbored an additional mutation, in *aceE* and *glnS*, respectively. By contrast, the remaining 13 suppressor mutations did not appear in more than one mutant. Currently, we cannot explain clearly how these mutations contribute to the L-alanine prototrophy in each suppressor mutant (#33-1, #44-1, #51, #64-1, #65-2, and #68-1) that did not also concurrently possess a mutation in *pta* or *aceE*. Intriguingly, however, among these mutants, three mutants (#64-1, #68-1, and #44-1) possess the aforementioned *cyaA* point mutation (C1957T). This mutation is expected to lower adenylate cyclase activity and thereby lead to reduced levels of the intracellular messenger cAMP, which, in turn, could regulate an as yet unknown pathway(s) and cause the accumulation of intracellular pyruvate. This possibility is in accord with the notion that the ratio [phosphoenolpyruvate]/[pyruvate] is positively correlated with the phosphorylation level of the phosphotransferase system component enzyme IIAGlc [34], and that the phosphorylated enzyme IIAGlc activates adenylate cyclase, which results in the production of cAMP [35].

To obtain more insight into the aforementioned complicated metabolic network, we must perform more in-depth experiments, such as the construction of the isogenic mutant harboring the *cyaA* point mutation (C1957T). In addition, we should address the important question of how L-alanine starvation stress induces mutation(s) by isolating mutants that do not show high-frequency of the emergence of L-alanine prototrophic suppressor mutants.

The most notable feature of the results obtained in this study is that the L-alanine prototrophic suppressors were generated by stress-induced mutagenesis. Bacteria constantly face environmental changes, which could be harmful under a given condition but also nonlethal under another. Bacteria respond to these stresses through mutation, where a cell that obtains a beneficial mutation survives to generate a mutant strain under the stress conditions. Mutants have long been recognized to arise through spontaneous mutation, which occurs without the influence of selective stress, as revealed by the elegant experiments of Luria and Delbruck [36] and Lederberg and Lederberg [37]. However, this concept has been challenged by the 1988 Cairns study of adaptive mutagenesis [38], which posits that mutations emerge after cells encounter a growth-limiting environment (i.e., a stress) instead of occurring independently of the stress. To explain this phenomenon, the hypermutable state model has been proposed, in which a cell subpopulation enters a transient hypermutation state that provokes a high mutation rate under stress conditions [39,40], and this process has further been shown to generate genome-wide mutation in the subpopulation [41]. This model could explain the occurrence of the L-alanine prototrophic suppressors obtained in this study, because genome sequencing of the independent suppressor mutants revealed that 14 genes, besides the genes related to the PDH complex and PTA-ACKA pathway, carried a mutation as described above, and more importantly, eight and one suppressor clones respectively harbored two and three distinct point mutations in different genes concurrently (Table 3 and Appendix A). Therefore, the emergence of L-alanine prototrophic suppressor mutants derived from the L-alanine auxotroph represents a favorable experimental system for a comprehensive investigation of stress-induced mutagenesis. Moreover, this study provides, to the best of our knowledge, the first example indicating that stress-induced mutagenesis can be used to select for a high-producer strain of a metabolite in the bioengineering process.

## Figures and Tables

**Figure 1 microorganisms-09-00472-f001:**
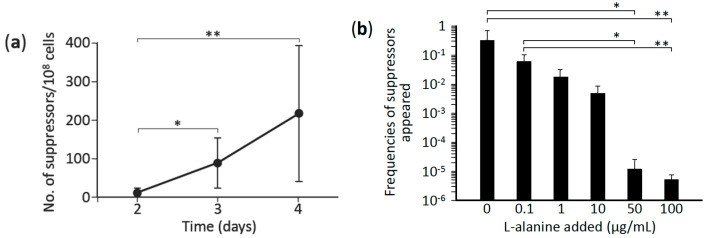
Effect of incubation time and L-alanine concentration on the appearance of L-alanine prototrophic suppressors. L-alanine auxotroph HYE032 cells were seeded on minimal agar medium without L-alanine supplementation and incubated at 37 °C for 2–4 days. (**a**) Colonies that newly appeared on each day were counted and the frequency of suppressors per 10^8^ cells inoculated was calculated. Points represent means and standard deviation from >3 independent experiments. (**b**) HYE032 cells were inoculated into a minimal liquid medium containing various concentrations of L-alanine (0.1–100 μg/mL) and incubated to late-log phase, and the frequency at which suppressors appeared was evaluated. Data represent means and standard deviation from 3 independent experiments. Statistical analyses; (**a**) * *p* < 0.05, ** *p* < 0.0001 and (**b**) * *p* < 0.01, ** *p* < 0.001.

**Figure 2 microorganisms-09-00472-f002:**
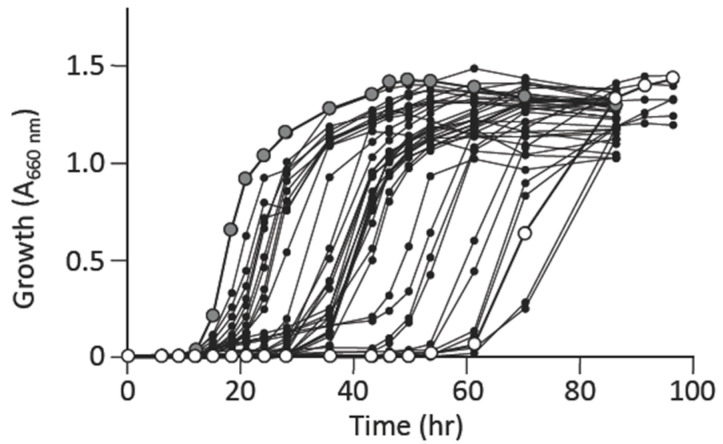
Growth of distinct L-alanine prototrophic suppressors that appeared on minimal agar medium. Here, 32 independent clones were grown in a minimal liquid medium without L-alanine supplementation. Symbols: large grey circles, W3110; large open circles, HYE032; small black circles, independent L-alanine suppressors. The figure shows a representative result from >2 experiments.

**Figure 3 microorganisms-09-00472-f003:**
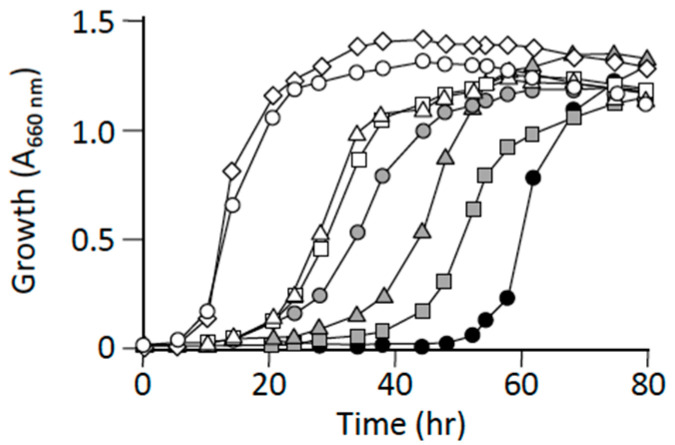
Effect of pyruvate on the growth of L-alanine auxotroph HYE032. HYE032 cells were inoculated into a minimal liquid medium containing various amounts of pyruvate (10–500 μg/mL), and growth was monitored by measuring the absorbance at 660 nm. Symbols: open diamonds, W3110 with no supplementation; open circles, HYE032 with 100 μg/mL L-alanine; open triangles, HYE032 with 500 μg/mL pyruvate; open squares, HYE032 with 300 μg/mL pyruvate; grey circles, HYE032 with 100 μg/mL pyruvate; grey triangles, HYE032 with 20 μg/mL pyruvate; grey squares, HYE032 with 10 μg/mL pyruvate; closed circles, HYE032 with no supplementation. The figure shows representative results from >2 experiments.

**Figure 4 microorganisms-09-00472-f004:**
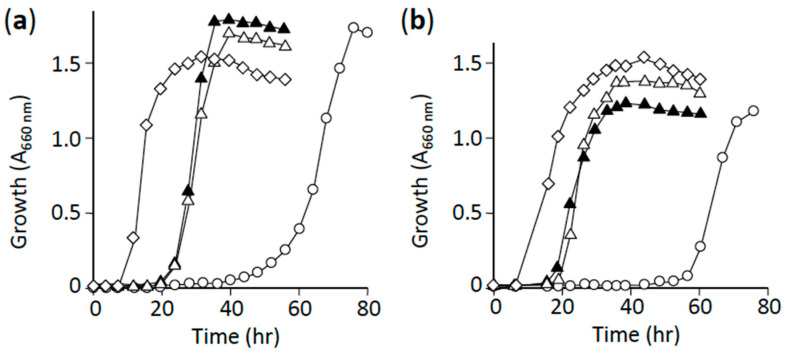
Growth of L-alanine prototrophic suppressor mutants and their cognate isogenic mutants in liquid minimal medium. (**a**) Clone #8-2 and its isogenic mutant #8-2M. (**b**) Clone #13-1 and its isogenic mutant #13-1M. Symbols: open diamonds, W3110; open circles, HYE032; open triangles, original L-alanine suppressors #8-2 (**a**) and #13-1 (**b**); closed triangles, respective isogenic mutants #8-2M (**a**) and #13-1M (**b**). The figure shows representative results from 3 independent experiments.

**Figure 5 microorganisms-09-00472-f005:**
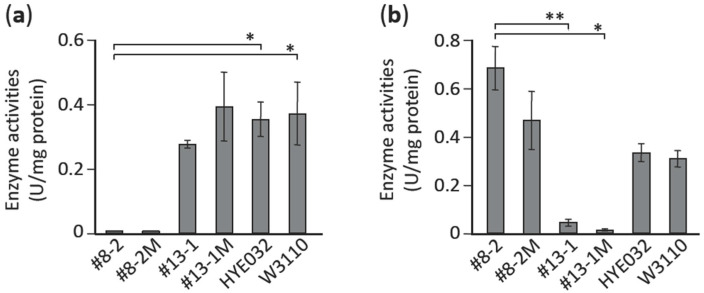
Enzyme activities of pyruvate transacetylase (**a**) and pyruvate dehydrogenase (**b**) in L-alanine suppressors #8-2 and #13-1 and their cognate isogenic mutants #8-2M and #13-1M. Data represent means and standard deviation of 3 independent experiments. Statistical analyses in (**a**) * *p* < 0.05 and (**b**) * *p* < 0.01, ** *p* < 0.001.

**Figure 6 microorganisms-09-00472-f006:**
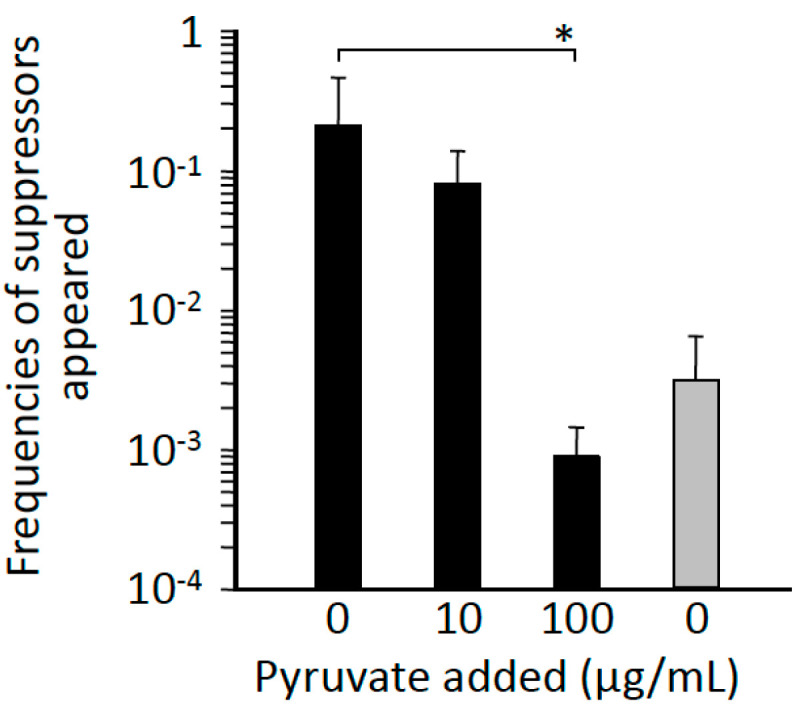
Effects of culture conditions on appearance of L-alanine prototrophic suppressors in minimal medium. HYE032 cells were inoculated into the minimal liquid medium and grown aerobically (black bars) in the presence of pyruvate (10 and 100 μg/mL) or grown anaerobically to the late-log phase (grey bar). Subsequently, the frequency of appearance of suppressors was measured. Data represent means and standard deviation from 3 independent experiments. Statistical analyses; * *p* < 0.05.

**Figure 7 microorganisms-09-00472-f007:**
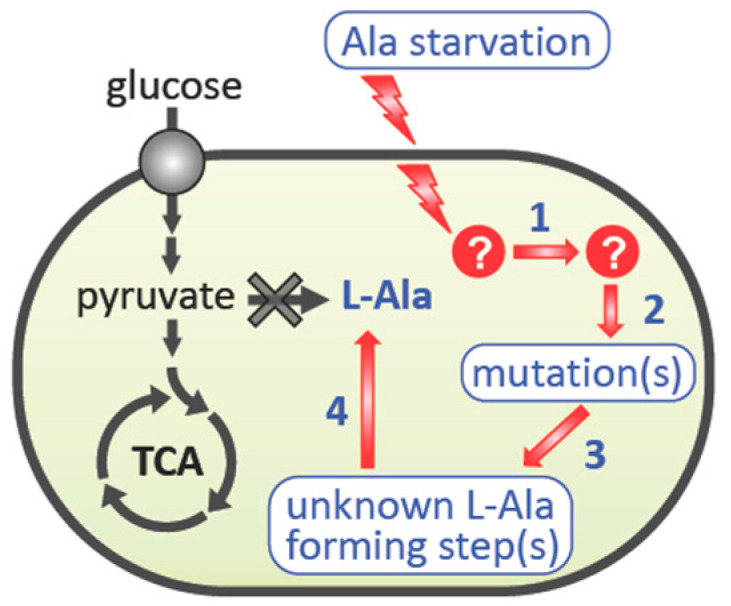
Model depicting stress-induced mutagenesis leading to the emergence of L-alanine prototrophic suppressor mutants. The L-alanine auxotrophic strain HYE032 senses the stress of L-alanine starvation, which leads to activation of a DNA-related metabolic pathway (Step 1). Subsequently, a genome-wide mutation(s) is introduced (Step 2) in a subpopulation of the cells. Some of the clones obtain, by chance, the ability to increase their intracellular level of pyruvate (Step 3), which provides HYE032 cells with an adequate amount of L-alanine to proliferate (Step 4).

**Table 1 microorganisms-09-00472-t001:** Strains and plasmids used in this study.

Strain or Plasmid	Relevant Properties	Source
**Strain**		
*Escherichia coli* W3110	Wild-type	Laboratory strain
HYE032	*avtA*::GM, *yfbQ*::KM, *yfdZ*::FRT, L-alanine auxotroph	[6]
JM109	*recA1*, *endA1*, *gyrA96*, *thi-1*, *hsdR17* (r_k_^−^m_k_^+^), *e14*^−^(*mcrA*^−^), *supE44*, *relA1*, Δ (*lac*-*proAB*)/F’[*traD36*, *proAB*^+^, *lacl*^q^, *lacZ* Δ M15]	Laboratory strain
#8-2M	Isogenic mutant of HYE032 with a point-mutation found in a suppressor mutant #8-2 in *pta* gene	This study
#13-1M	Isogenic mutant of HYE032 with a point-mutation found in a suppressor mutant #13-1 in *aceE* gene	This study
#3 to #70 *	L-alanine prototrophic derivatives isolated from HYE032	This study
**Plasmid**		
pTH18cs1	*cat1*, *lacZ’*, *repA_ts1_*, derivative of pSC101	[15]
pTH8-2	pTH18cs1 harboring 0.5-kb PCR fragment of *pta* gene	This study
pTH13-1	pTH18cs1 harboring 1.8-kb PCR fragment of *aceE* gene	This study

* Detailed information on L-alanine prototrophic derivatives is listed in Table 3.

**Table 2 microorganisms-09-00472-t002:** Primers used for the construction of isogenic mutants.

Primer	Nucleotide Sequence (5′-3′)	Comment
pta-Fwd	tgtctagaCACTACCGCAAACACCATCC	*Xba*I tag *
pta-Rev	gttctagaACCAACGTATCGGGCATTGC	*Xba*I tag *
aceE-Fwd	attctagaATTCGCGTCGCAATTGCTCT	*Xba*I tag *
aceE-Rev	tttctagaAACGAAAGCTTCAGAGCACG	*Xba*I tag *
pta-seq	GATGACGAGATTACTGCTGC	for sequencing
aceE-seq	AAAAGACCTCGAACTGGGC	for sequencing

* Lowercase letters including underlined nucleotides represent tags used for cloning.

**Table 3 microorganisms-09-00472-t003:** Results of mutations in individual L-alanine prototrophic suppressor mutants.

Suppressor Strain	Days of Isolation	Gene	Nucleotide Change	Amino Acid Change
#3	2	*aceE*	T110A	I37N
		*flgJ*	A123 > AA *^1^	42 frameshift
#4-2	2	*aceE*	T110A	I37N
#5-1	2	*aceE*	A775G	N259D
#8-2	2	*pta*	A1967C	D656A
#10-1	2	*aceE*	T110A	I37N
		*cyaA*	C1957T	R653C
#11-2	2	*aceF*	A1504C	T502P
#12-2	2	*aceE*	A775C	N259H
#13-1	2	*aceE*	G440A	G147D
#14-1	2	*aceE*	C802T	P268S
		*cpxP*	T269G	F90C
		*yhbO*	T5A	S2R
#15-1	2	*pta*	A1967C	D656A
		*gspE*	A1381C	K461Q
#16-2	2	*pta*	A1967C	D656A
#17-2	2	*pta*	A1967C	D656A
		*plsB*	G2198T	R733L
#18-1	2	*pta*	A1967C	D656A
#19	2	*pta*	A1967C	D656A
#20	2	*pta*	G1420T	E474stop
#21	2	*pta*	A1967C	D656A
#23	2	*pta*	A1967C	D656A
		*ydhK*	C305A	A102E
#24	2	*pta*	T2A*^2^	No initiation
#25	2	*pta*	A1967C	D656A
#27	2	*pta*	A1967C	D656A
#29	2	*pta*	A1967C	D656A
#31-1	3	*lpd*	C963A	H322Q
#32-1	3	*aceE*	G1193T	G398V
		*yhiX*	C831T *^3^	Silent
#33-1	3	*entE*	C827T	T276I
#35-1	3	*pta*	A1967C	D656A
#37-1	3	*aceF*	C1700A	A567E
#39-2	3	*aceE*	C617T	A206V
		*aceE*	G810A *^3^	silent
#40-2	3	*pta*	C1884G	D628E
#43-1	3	*lpd*	T956G	L320R
#44-1	3	*cyaA*	C1957T	R653C
#45-1	3	*pta*	A1967C	D656A
#46-2	3	*pta*	A1967C	D656A
#48-1	3	*aceE*	G440A	G147D
		*flhC*	C115 deletion *^4^	39 frameshift
#51	3	*ynbB*	A380C	D127A
#54	3	*aceE*	T584C	M195T
		*ftsI*	C1599T*^3^	Silent
#57	3	*lpd*	T956G	L320R
#64-1	4	*cyaA*	C1957T	R653C
#65-2	4	*yhjG*	T543A	D181E
#68-1	4	*cyaA*	C1957T	R653C
		*glnS*	A956T	Q319L
#70	4	*ackA*	A878 deletion *^4^	293 frameshift

Amino acid residues are represented using the one-letter code. *^1^ Adenine is inserted after 123A, causing a frameshift mutation at amino acid residue 42. *^2^ T2A mutation disrupts the initiation codon. *^3^ Nucleotide changes in suppressor mutants #32-1, #39-2, and #54 generate silent mutations. *^4^ Deletion of a nucleotide in suppressor mutants #48-1 and #70 causes a frameshift mutation at amino acid residues 39 and 293, respectively.

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
