# Peer review of "L-Alanine Prototrophic Suppressors Emerge from L-Alanine Auxotroph through Stress-Induced Mutagenesis in Escherichia coli"

_microorganisms, 2021, doi:10.3390/microorganisms9030472_

Round 1
Reviewer 1 Report
In the present study entitled “L-Alanine Prototrophic Suppressors from L-Alanine Auxotroph Emerge through Stress-Induced Mutagenesis in Escherichia coli”, the Authors have examined the appearance, at relatively high frequency, of L-alanine prototrophic mutants. These mutants, although lacking the three isozymes responsible of the synthesis of alanine, acquired the ability to grow in a minimal medium not supplemented with this amino acid. Whole-genome analysis of 40 L-alanine prototrophic mutants show that most of mutations map in pta and aceE genes that control the intracellular level of piruvate. Importantly, the Authors demonstrate that point mutations in pta (D656A) and aceE (G147D) confer the alanine prototrophic phenotype to the HYE032 cells. Accordingly, the addition of piruvate to the growth medium abolishes the requirement for alanine of the L-alanine auxotroph HYE032 strain.
The Authors propose a very interesting hypothesis to explain their results.
The pyruvate accumulation affects the cell metabolism and under this condition, secondary enzymes, not usually synthesizing alanine, acquire this new function. A general concern could be that these not canonical enzymes for the biosynthesis of alanine in suppressor mutants have not been identified.
Although this study is not definitive, the strategy is well designed and experiments support the main conclusions drawn by the authors.
Specific comments
- Results from supplementary Fig. S1b and Fig. 2 seem contradictory. In Fig. 1Sb suppressor mutants show comparable growth rates whereas differences in growth are clearly evident in Figures 2 and 2S. The text of paragraph 3.3 is quite confusing (particularly the premise and conclusions) and should be largely reviewed.
- It is reasonably that a high amount of piruvate could cause a further decrease of the number of alanine suppressor mutants also under anaerobic growth. In my opinion, this control should be added to Fig. 6.
- Concentrations reported as mmol/L should be indicated with "mM".
- Figure 6 is misreported in line 387.
Author Response
Thank you for your insightful and constructive comments that helped improve our manuscript. We have addressed each comment and the point-by-point responses have been given below:
General comment:
We agree with your concern that this study is not definitive as secondary enzyme(s) involved in the biosynthesis of L-alanine in suppressor mutants have not yet been identified. As we pointed out in the manuscript (lines 59–63), the phenomenon of multicopy suppression has been observed in L-alanine auxotrophs in our previous study (ref 6), wherein the introduction of the wild-type serC gene into a multicopy plasmid in the L-alanine auxotroph HYE032 conferred a L-alanine prototrophic phenotype on the mutant cells. We therefore expected that a mutation leading to overexpression of the serC gene could be found through NGS analysis in the beginning, but we could not identify such a mutation(s). In addition, another study (ref 7) reported that, besides serC, other transaminase genes (argD, astC, aspC, gabT, puuE,tyrB, and ygiG) can suppress the growth defect of the L-alanine auxotroph. Thus, it is reasonable to speculate that several of these enzymes (single or in some combination) could be involved in L-alanine formation once the intracellular pyruvate levels are increased.
Specific comment 1:
Thank you for the comment and suggestions that helped improve our manuscript. Fig. S1 and Fig. 2 are not contradictory. Individual clones shown in Fig. S1 were isolated from cells grown in minimal “liquid” medium in the absence of L-alanine, wherein HYE032 cells reached the late-log-phase. On the other hand, individual clones shown in Fig. 2 were isolated from cells grown on minimal “agar” medium without L-alanine supplementation. Thus, all clones grown on the “agar” medium are perfectly distinct suppressor clones, i.e. genetically independent clones. To clarify this point, we have revised paragraph 3.3 in the manuscript. Please see lines 216, 218, and 224–225.
Specific comment 2:
Thank you for your comment. We have added a description regarding the control experiment in the manuscript as per your suggestion. Please see lines 350–351.
Specific comment 3:
Thank you for your comment. We have edited the manuscript as per your suggestion. Please see lines 82, 83, 142, 152, 153, 155, 158, 159, and 160.
Specific comment 4:
Thank you for your comment. This was a typographical mistake. We have changed “Figure 6” to “Figure 5”.

Reviewer 2 Report
Harutaka Mishima et al provide an interesting description of L-alanine prototrophic mutants from L-alanine auxotroph HYE032 and suggest a stress-induced mechanisms. The methods are adequately described and results clearly explained.
Author Response
Thank you very much for taking the time to review our manuscript “L-Alanine Prototrophic Suppressors from L-Alanine Auxotroph Emerge through Stress-Induced Mutagenesis in Escherichia coli” (manuscript ID: microorganisms-1105662). We are glad to receive your positive comments.